# Tropical Atlantic Mixed Layer Buoyancy Seasonality: Atmospheric and Oceanic Physical Processes Contributions

**Ibrahima Camara [1],\*, Juliette Mignot [2], Nicolas Kolodziejczyk [3], Teresa Losada [4] and Alban Lazar [2]**

[1] Laboratoire de Physique de l'Atmosphère et de l'Océan (LPAO), ESP/UCAD/, Dakar 5085, Senegal
[2] LOCEAN Laboratory, Sorbonne Universités (CNRS/IRD/MNHN), 75005 Paris, France; Juliette.Mignot@locean-ipsl.upmc.fr (J.M.); alban.lazar@locean-ipsl.upmc.fr (A.L.)
[3] UBO, CNRS, IRD, Ifremer, Laboratoire d'Océanographie Physique et Spatiale, 29280 Plouzané, France; nicolas.kolodziejczyk@univ-brest.fr
[4] Departamento de Física de la Tierra y Astrofísica, Universidad Complutense de Madrid, 28012 Madrid, Spain; tldoval@fis.ucm.es
**\*** Correspondence: ibrahima1.camara@ucad.edu.sn

**Abstract:** This study investigates the physical processes controlling the mixed layer buoyancy using a regional configuration of an ocean general circulation model. Processes are quantified by using a linearized equation of state, a mixed-layer heat, and a salt budget. Model results correctly reproduce the observed seasonal near-surface density tendencies. The results indicate that the heat flux is located poleward of 10° of latitude, which is at least three times greater than the freshwater flux that mainly controls mixed layer buoyancy. During boreal spring-summer of each hemisphere, the freshwater flux partly compensates the heat flux in terms of buoyancy loss while, during the fall-winter, they act together. Under the seasonal march of the Inter-tropical Convergence Zone and in coastal areas affected by the river, the contribution of ocean processes on the upper density becomes important. Along the north Brazilian coast and the Gulf of Guinea, horizontal and vertical processes involving salinity are the main contributors to an upper water change with a contribution of at least twice as much the temperature. At the equator and along the Senegal-Mauritanian coast, vertical processes are the major oceanic contributors. This is mainly due to the vertical gradient of temperature at the mixed layer base in the equator while the salinity one dominates along the Senegal-Mauritania coast.

**Keywords:** physical processes; salt and heat budget; density; compensation

## 1. Introduction

In the climate system, the ocean and atmosphere interact by exchanging momentum, heat, freshwater, and various tracers such as carbon dioxide and oxygen. In the upper ocean, these exchanges take place in the mixed layer. The latter is characterized by roughly vertically homogeneous profiles of temperature and salinity and, thus, density. The thickness of the mixed layer largely determines the amount of heat available for the air-sea interaction as well as the potential energy associated with upper ocean dynamics [1]. In the tropics, it is generally the temperature that controls the stratification and the mixed layer thickness. Yet, in regions of a strong vertical salinity gradient in the upper ocean, haline stratification can be strong enough to compete with the temperature and even eventually control the density stratification. This haline stratification is controlled by precipitation and/or runoff as well as lateral advection both at the surface and in the subsurface. In the tropical Atlantic, intense precipitation and runoffs are the most common factors strongly influencing the vertical salinity gradient in the upper

ocean. When the density stratification is controlled by salinity, an isothermal layer can be found below the mixed layer. It is called a barrier layer, and it prevents exchanges between the warm mixed layer and the cold ocean interior [2,3]. This barrier layer can have a strong impact on oceanic circulations [4].

Additionally, the mixed layer density can impact the exchanges between the surface and water masses of the interior ocean. Winter convection in some areas contributes to the mixing of surface and subsurface water and generates new water masses [5–7]. When the upper ocean re-stratifies in the spring, the mixed layer shallows and, water masses leave the turbulent mixed layer to enter the interior ocean by detrainment, and it roughly follows the isopycnal layers, connecting surface properties to interior water masses [8–10]. For example, each spring, 13 Sv (1 Sv = $10^6$ m$^3$/s) of water newly formed within the South Atlantic mixed layer is exported toward the Indian ocean before subducting [11]. Hence, understanding and quantifying regional variations of seasonal mixed layer density are of wide interest for evaluating the ocean interior.

Using Argo data, Reference [12] outlined the relative contribution of temperature and salinity on density in the upper global ocean. The researchers found that, as indicated above, temperature changes generally dominate the seasonal cycle of mixed layer density in tropical to mid-latitude regions. Salinity changes are, nevertheless, dominant in the tropical warm pools as well as in polar regions (Arctic and Austral ocean). This dominance is due to the effect of intense precipitation, river runoffs, and ice melting. In the tropical ocean, under the Inter-tropical Convergence Zone (ITCZ), they found that the upper ocean temperature and salinity both contribute to increase the stratification. In the eastern subtropics, on the other hand, the effect of seasonal salinity variations on density compensates the first temperature, which leads to weak mixed layer density variation [5,12,13]. In the tropical Atlantic, using salinity and temperature from the World Ocean Database 2005 of the National Oceanographic Data Center [14,15], it was found that the potential density increases from 0 to 700 m, which is due to salinity. Lastly, Reference [16] quantified that, in many regions of the tropics, the ocean salinity stratification contributes to 40–50% of the total stratification as compared to the thermal stratification and, in some specific regions, the former exceeds the latter for a few months of the seasonal cycle.

These studies contributed to diagnose the relative contribution of temperature and salinity to the density. However, they did not investigate the relative contributions of the individual physical processes of the temperature and salinity budget into the buoyancy budget. On the surface, both heat and freshwater fluxes, i.e., buoyancy fluxes, impact the water masses transformations [6,17–19]. On the subsurface, horizontal and vertical mixing of salinity and temperature, i.e., buoyancy mixing, also impact mixed layer water [20,21].

This study aims at better understanding the physical mechanisms that drive variations of the oceanic upper density. For this, we use a regional configuration forced by reconstructed atmospheric fields of the NEMO-OPA (Ocean Parallélisé) Ocean General Circulation Model (OGCM) in the tropical Atlantic to quantify the relative contribution of oceanic and atmospheric physical processes entering in the heat and salt budget as well as the density budget. The model and methodology are described in Section 2. The results are presented in Section 3. We will first present a validation of the mean modeled density and its linear decomposition into temperature and salinity contribution. Second, an overview of the mean heat and salt physical processes' relative contribution on density is proposed. Third, we consider the seasonal modulation of these contributions. Summary and discussions are presented in Section 4.

## 2. Model and Methodology

### 2.1. Model Description

This study is based on an oceanic model NEMO-OPA [22] forced by reconstructed atmospheric fields. OPA solves the primitive equations on an Arakawa C grid with a second order finite difference scheme, assuming the business and hydrostatic approximations, the incompressibility hypothesis,

and free surface formulation [23]. The density is computed from potential temperature, salinity, and pressure, using the equation of state from Reference [24]. The horizontal mesh is based on a 0.25° × 0.25° Mercator grid. We use the regional configuration of the tropical Atlantic, named ATLTROP025. The grid has been limited to 30° N and 30° S. Realistic bottom topography and coastline are derived from ETOPO2. The maximum depth of 5000 m is spanned by 46 z-level ranging from 5 m thickness in the upper 30 m and 10 m thickness around 100 m depth. The ocean model is run with a time step of 1440 s. Outputs are provided over 5 day-averages. Ocean boundary conditions are taken from the output of a global $\frac{1}{4}°$ simulation part of the DRAKKAR hierarchy of global configurations [25] and detailed in Reference [26]. Surface boundary forcing is ensured by bulk formulas developed by Reference [13]. The momentum fluxes (latent and sensible fluxes) were computed with the surface 6-hourly atmospheric state variables (air temperature, humidity, and winds at 10 m). Then, we forced the model for 13 years with climatological weekly fields averaged over the period 1988–2000. For the present study, forcing data were taken from global DRAKKAR forcing set 4 (DFS4) [26]. In this configuration, a Newtonian restoring term was added into the salinity equation (Equation (3) below). This restoring is expressed as: $-\gamma(S–S_0)$, where $\gamma$ is the relaxation time scale (33 days for the first 10 m) and $S_0$ is a given salinity (usually a climatology).

### 2.2. Density as a Function of Temperature and Salinity

The equation of the state is non-linear and it depends on temperature, salinity, and pressure [24]. In the model, the seawater density is computed diagonally from temperature, salinity, and pressure using the UNESCO 1983 [27] polynomial equation of state. At regional scale, if the salinity and temperature spatial and temporal variations are not too large, one can linearize the density equation around an average value of temperature and salinity. This yields the following linearized equation of state.

$$\rho_{ml} - \rho_0 = \rho_0(\beta[S_{ml} - S_0] - \alpha[T_{ml} - T_0]) \tag{1}$$

where $\rho_{ml}$ is the vertically averaged density of seawater in the mixed layer (*ml*), expressed in kg.m$^{-3}$. $\rho_0 = 1025.98$ kg.m$^{-3}$ is the reference density at $S_0 = 35$ psu, $T_0 = 15$ °C. $\beta$ (in psu$^{-1}$) and $\alpha$ (in °C$^{-1}$) are the haline contraction and thermal expansion coefficient, respectively. They depend on temperature and salinity. As a first approximation, in this case, they are considered at the surface pressure. In this study, they are computed posteriori using a polynomial function of temperature and salinity. If the temporal variation of $\alpha$ and $\beta$ are furthermore neglected, the temporal derivation of Equation (1) gives the following.

$$\frac{\partial \rho_{ml}}{\partial t} = \rho_0\left(\beta\frac{\partial S_{ml}}{\partial t} - \alpha\frac{\partial T_{ml}}{\partial t}\right) \tag{2}$$

with $\frac{\partial S_{ml}}{\partial t}$ and $\frac{\partial T_{ml}}{\partial t}$ being the salt and heat budget tendencies, respectively, within the mixed layer. Both approximations made above were verified in the model and in validation data. The salt and heat budget tendencies can, in turn, be written as follows.

$$\frac{\partial S_{ml}}{\partial t} = \underbrace{-\frac{1}{h}\int_{-h}^{0}u\partial_xSdz - \frac{1}{h}\int_{-h}^{0}v\partial_ySdz - \frac{1}{h}\int_{-h}^{0}D_l(S)}_{OCEHOR_S} \overbrace{\qquad\qquad\qquad\qquad\qquad}^{OCE_S} \underbrace{-\frac{1}{h}(S_{ml} - S_{-h})(w_{-h} - \partial_th) - \frac{1}{h}[k_z\partial_zS]_{-h}}_{OCEVER_S} + \underbrace{\frac{E - P - R}{h}S}_{ATM_S} \tag{3}$$

$$\frac{\partial T_{ml}}{\partial t} = \underbrace{-\frac{1}{h}\int_{-h}^{0}u\partial_xTdz - \frac{1}{h}\int_{-h}^{0}v\partial_yTdz - \frac{1}{h}\int_{-h}^{0}D_l(T)}_{OCEHOR_T} \overbrace{\qquad\qquad\qquad\qquad\qquad}^{OCE_T} \underbrace{-\frac{1}{h}(T_{ml} - T_{-h})(w_{-h} - \partial_th) - \frac{1}{h}[k_z\partial_zT]_{-h}}_{OCEVER_T} + \underbrace{\frac{Q_s(1 - F_h) + Q_{ns}}{\rho_0C_ph}S}_{ATM_T} \tag{4}$$

where $T_{ml}$ and $S_{ml}$ are, respectively, the vertically averaged temperature and salinity in the mixed layer (depending on longitude, latitude, and time). *T* and *S* are temperature and salinity, respectively, depending on the 3-dimensional space and time. The *u*, *v*, and *w* stand for the oceanic current in the zonal, the meridional, and vertical direction, respectively, defined at each model's vertical level. $T_{-h}$ and $S_{-h}$ are the temperature and salinity at the base of the mixed layer, $k_z$ is the vertical diffusion coefficient of tracers, $D_l$ is the horizontal diffusion operator, and h is the depth of the mixed layer, computed using a density criterion. It corresponds to the depth at which the density has increased by 0.01 kg.m$^{-3}$ from the value at the first vertical level of the model surface value. In Equation (3), *E-P-R* is the surface freshwater flux including evaporation (*E*), precipitation (*P*), and river runoff (*R*). In Equation (4), $Q_{ns}$ and $Q_s$ are the nonsolar and solar components of the air-sea heat flux and $F_h$ is the fraction of the shortwave radiation that reaches the mixed layer depth where $\rho_0$ is the water density and Cp is the specific heat. In Equation (3), we summarize each of the terms of the budget by an explicit acronym, which allows distinguishing atmospheric ($ATM_S$) from oceanic ($OCE_S$) effects involving salinity and then vertical ($OCEVER_S$) and horizontal ($OCEHOR_S$) processes in the ocean. We proceed identically in Equation (4) for the terms involved in temperature changes. Such an approach based on either the salt budget or the heat budget have been used in several Atlantic studies based on observations [28–30] and models [17,31–33]. To our knowledge, the joint temperature and salinity budget approach has never been applied to assess the density budget in the upper ocean (only in the case of spiciness, see Reference [29]).

Multiplying each term of Equations (3) and (4) by $(\rho * \beta)$ and $(\rho * \alpha)$, respectively, allows scaling all terms in density units (kg.m$^{-3}$) and, thereby, allows comparing the respective weight of haline and thermal terms on density. In the following, we will consider.

$$\text{ATM}_\rho = \rho * \beta * ATM_S - \rho * \alpha * ATM_T \tag{5}$$

The contribution of atmospheric physical processes on mixed layer density is expressed in terms of an increase of oceanic density when the flux is positive. It depends on the freshwater flux and air sea heat flux contribution to the buoyancy flux (expressed $ATM_{S\rho} = \rho * \beta * ATM_S$ and $ATM_{T\rho} = -\rho * \alpha * ATM_T$, respectively).

We will similarly consider

$$\text{OCE}_\rho = \rho * \beta * OCE_S - \rho * \alpha * OCE_T \tag{6}$$

the contribution of oceanic physical processes on mixed layer density, which can be derived in a thermal ($OCE_{T\rho} = -\rho * \alpha * OCE_T$) and haline ($OCE_{S\rho} = \rho * \beta * OCE_S$) part.

The contributions of each of these physical terms to the mixed layer density are analyzed in Section 3.

### 2.3. Mixed Layer Density: Model-Observation Comparison and Linearization

Since this study focuses on the density seasonal variations, we first compare the model densities with observed density through its seasonal variations, which are known as named density tendencies. For this, we use Sea Surface Salinity (SSS) from Soil Moisture Ocean Salinity (SMOS) and Optimal Interpolation Sea Surface Temperature (OISST) [34,35]. We also computed the density averaged in the mixed layer at the monthly time scale from both SODA reanalysis and EN4 dataset, and compared it with the first level (5 m) density in the same re-analysis (not shown). We found no important difference between these two fields, which suggested that the surface density is very close to the density averaged in the mixed layer depth and the former can, thus, be used to validate the latter. For this reason, we decided to compare the model results to surface data from observations, which are more accurate than re-analysis data. The OISSTs used here are daily products obtained by using only microwave data at 0.25° × 0.25° resolution. The L3 SMOS SSS v3 data are produced at C Expert Center (CEC) LOCEAN (Paris) and are corrected from systematic errors by preserving their temporal dynamic [36,37].



Although the actual SMOS resolution is about 45 km and the revisit time is 4 days, the CEC SSS products are provided every 4 days from 01/2010 to 12/2017 and are gridded on a 0.25° × 0.25°. The ocean density tendency is computed from the UNESCO 1983 (EOS 1980) polynomial density equation both in the model and in the data. A comparison between April-August (known as boreal summer) and October-February (known as boreal winter) periods are shown in Figure 1a,b, respectively. These sets of months were chosen for spatio-temporal coherency in order to maximize the density tendency that basically keeps the same sign over these periods locally (not shown). During boreal summer, the density primarily decreases north of approximately 5° N and decreases to the south (Figure 1a). An opposite pattern characterizes a boreal winter (Figure 1b). The model patterns are in good agreement with the model data (Figure 1c,d) especially in the open ocean far from the coast. Around the main rivers mouth of the basin (Amazon, Congo, and Senegal river), the model, nevertheless, overestimates the mixed layer density variation. The model also overestimates the upper layer density variation in the deep tropics under the path of the seasonal march of the ITCZ during the boreal summer. These overestimations could also be due to a salinity restoring term that can strongly affect the oceanic mean state [38,39]. The basin scale signals are primarily explained by the temperature seasonal cycle (not shown). Contribution from salinity variations are more local and associated with freshwater inputs from the ITCZ and rivers run-off off North Brazil and in the Gulf of Guinea [17].

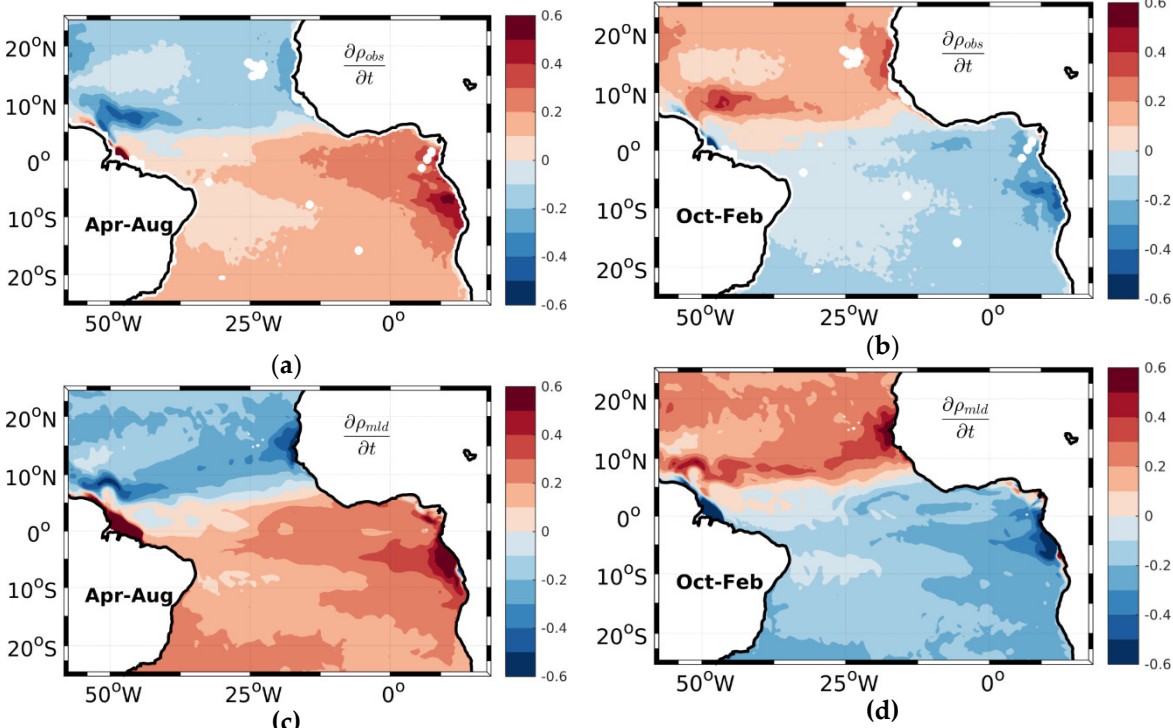

**Figure 1.** April-August (**a**) and October-February (**b**) average of surface density tendency from observations. Observational data are Sea Surface Salinity (SSS) from Soil Moisture Ocean Salinity (SMOS) and Optimal Interpolation Sea Surface Temperature (OISST) obtained by using only microwave data. (**c**,**d**) are the same as (**a**,**b**) for the density averaged over the mixed layer depth in the simulation. Units are in $kg.m^{-3}month^{-1}$.

In order to follow the decomposition proposed in Section 2.2, one must also validate the linear approximation proposed in Equation (1) as well as the fact that temporal variations of $\alpha$ and $\beta$ can be neglected. Figure 2 shows the density tendency obtained from Equation (2) for both data and the model. The linearized density equation (Figure 2) is in good agreement with the UNESCO 1983 polynomial one (Figure 1) especially for observational data (Figure 2). As in Figure 1, we find strong variations around river mouths and also in the seasonal march of the ITCZ. In the model, the spatial distribution

in the open ocean differs slightly between a linear and nonlinear form (Figure 1c,d, and Figure 2c,d). The amplitude of the linear form is also overestimated in the Senegal-Mauritania upwelling area. The matching between the linearized equation and the polynomial one justifies the use of the linearized version to reconstruct the upper density in the rest of this study.

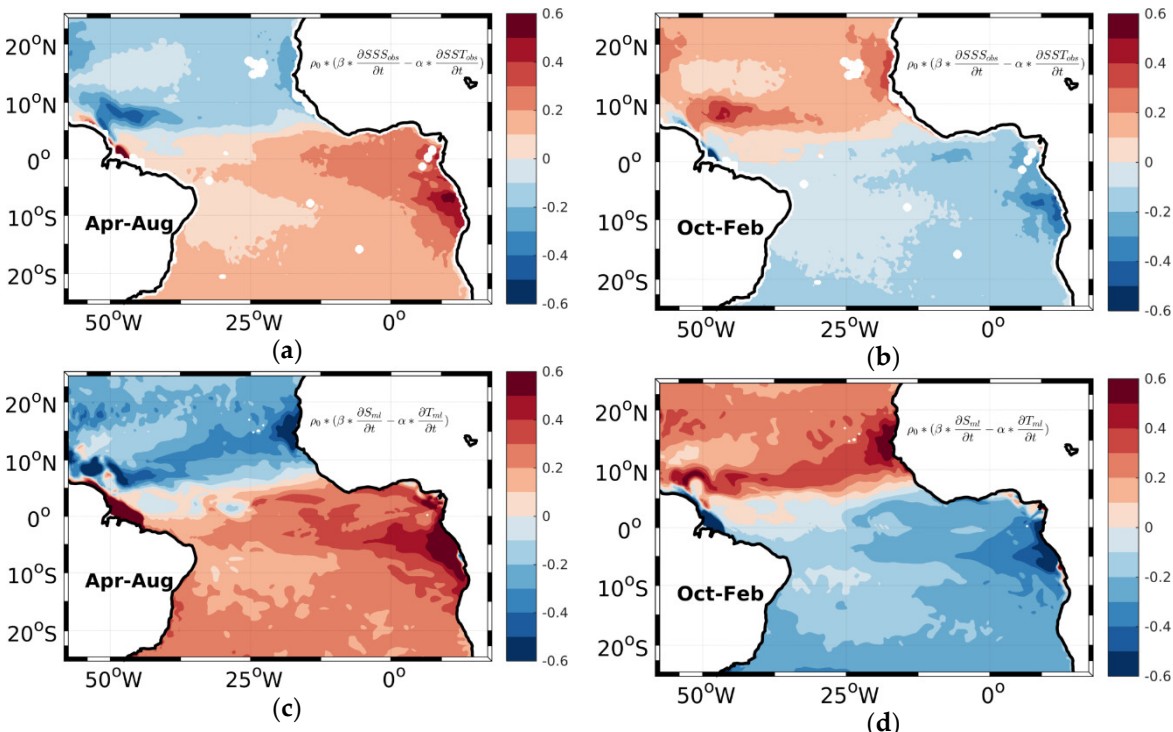

**Figure 2.** April-August (**a**) and October-February (**b**) average of surface density tendency from observations. The density is obtained with a linear density equation. (**c**,**d**) are the same as (**a**,**b**) for model data. Units are in kg.m$^{-3}$month$^{-1}$.

## 3. Results

### 3.1. Atmospheric Physical Processes Contribution

Figure 3a,b (shaded) show, for boreal summer and winter, respectively, the time-mean average of the atmospheric physical processes (noted ) contribution to mixed layer density variations. During the boreal summer, the atmospheric fluxes tend to decrease the mixed layer density in most of the northern hemisphere while they increase it in the southern hemisphere. The situation is opposite for the boreal winter period. This result is consistent with a primary role of thermal fluxes, which warm the ocean in the local summer and cool it in the winter.

The dots in Figure 3a,b show regions where the atmospheric thermal (ATM$_{T\rho}$) and haline (ATM$_{S\rho}$) contributions to the ocean buoyancy flux act constructively (i.e., they have the same sign). During the boreal summer, both components generally act constructively in the southern hemisphere (poleward of 5° S, where it is winter locally) and destructively (i.e., they have an opposite sign) in the northern hemisphere poleward of 10° N. Again, the situation is opposite for the boreal winter. In the local summer, shortwave radiation is the major contributor to air-sea heat flux [32,40], which increase the mixed layer density while evaporation is the main component in freshwater fluxes. Hence, ATM$_{T\rho}$ and ATM$_{S\rho}$ do not have the same effect on buoyancy. Clearly, Figure 3a shows that the thermal effect dominates. In local winter, air-sea heat fluxes are controlled by latent heat fluxes, e.g., [40], which increase the upper density while freshwater is still dominated by evaporation [17] out of the ITCZ seasonal march. This explains ATM$_{T\rho}$ and ATM$_{S\rho}$ opposite effects on the buoyancy. Figure 3c,d show that the thermal term (ATM$_{T\rho}$) dominates by a factor of at least two ($\log_{10}\left(\frac{\alpha ATM_T}{\beta ATM_S}\right) > 0.3$) both

in the winter and summer in the extra-tropical regions. Along the south Brazilian coast, the factor is greater than 10.

This general picture is very different under the seasonal march of the ITCZ and along the coasts. In the western equatorial tropical Atlantic, the atmospheric freshwater and heat flux both induce a decrease of the upper ocean density all year long. The heat flux contribution is at least three times greater than the freshwater one. Immediately north of the equator, around 5° N, the two terms partly compensate each other all year long with a domination of freshwater flux. The latter is two times greater than the heat flux contribution to buoyancy (Figure 3c,d shaded). This is a signature of the seasonal cycle of the high precipitation and runoff under the ITCZ. Along the north Brazilian and West African coast, both terms act constructively throughout the year (Figure 3a,b black dots). During the boreal summer, the freshwater term is no more dominated by evaporation as in the open ocean but rather by precipitations and runoffs (Amazon and Senegal river, e.g., [17,40]). During the boreal winter, the decrease of the ocean buoyancy is due to latent heat flux and evaporation, which are the main components of air-sea heat and freshwater fluxes, respectively [17,40,41]. During the summer, the freshwater effects are greater than the shortwave radiation effect by a factor of two. However, during the winter, the latent heat flux is the major atmospheric contributor.

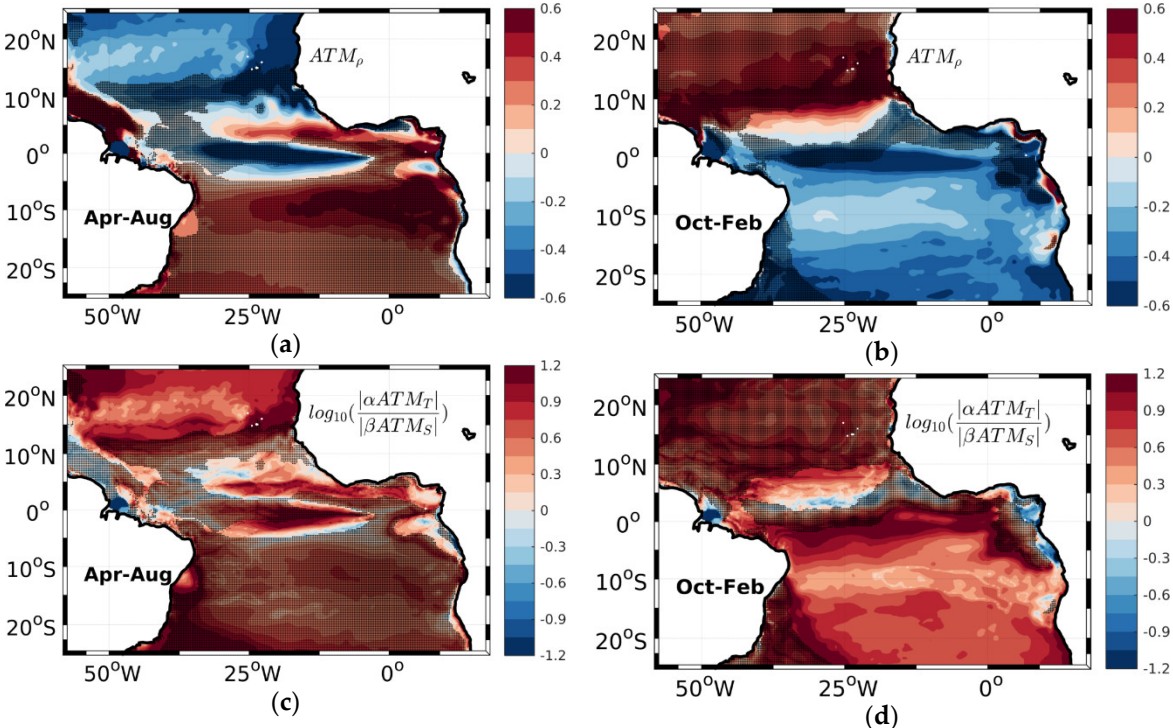

**Figure 3.** April-August (**a**) and October-February (**b**) average of the contribution of total atmospheric physical processes to mixed layer density variations. Units are in kg.m$^{-3}$month$^{-1}$. $ATM_\rho = \rho * \beta * ATM_S - \rho * \alpha * ATM_T$ (5). (**c**,**d**) are the logarithm of the ratio between the absolute value of the air-sea heat flux and freshwater flux contributions to the density variations. On all panels, the region where air-sea heat flux and freshwater flux contribution on mixed layer density is the same in terms of the signs indicated with black dots.

### 3.2. Oceanic Physical Processes Contribution

Figure 4 analyses the contribution of the oceanic processes to the mixed layer density (noted $OCE_\rho$). The general picture is very different from the emerging patterns of atmospheric fluxes. In this case, both seasons yield a similar, large-scale picture. Positive oceanic fluxes are found all year long in the central equatorial Atlantic and at the poleward edge of the subtropical gyres. Negative values are found around 10° to 15° north and south. The North Equatorial Counter Currant (NECC) seasonal retroflection

is one of the clear seasonal features of this oceanic contribution. During the boreal summer, it leads to a reduction of density in the west of the basin and extending eastward around 10° N. This reduction is primarily due to the haline contribution (Figure 4c), which dominates by a factor greater than two. Around 10° S, within 15° N–20° N and along the north Brazilian coast, $OCE_\rho$ diminishes the mixed layer density with a maximum density along the Brazilian coast. Thermal and haline processes act together in these three regions (Figure 4: black dots). In these areas, thermal processes heat the upper water and, at the same time, oceanic haline processes freshen the mixed layer regions [17,40]. Their relative contributions on density are furthermore of similar magnitude. The strongest increase of density (>0.6 kg.m$^{-3}$.month$^{-1}$) occurs in the western equatorial band. Thermal and haline oceanic processes act both to increase the mixed layer density (Figure 4a,b: black dots) due to cooling and the accumulation of salt by the equatorial and coastal upwelling [17,32,42]. In the equator, $OCE_{T\rho}$ is the major contributor with a factor of more than 4 (Figure 4c,d shaded, $\log_{10}\left(\frac{\alpha OCE_T}{\beta OCE_S}\right) > 0.6$). Poleward of 15° S, density also increases all year long through oceanic processes, but the thermal and haline effect have opposite signs with a larger contribution of the former with a factor of two mainly during Oct–Feb (Figure 4d: shaded). In this area, oceanic haline processes freshen the mixed layer [17] while thermal ones cool it [40]. Hence, the opposite effect occurs on oceanic buoyancy.

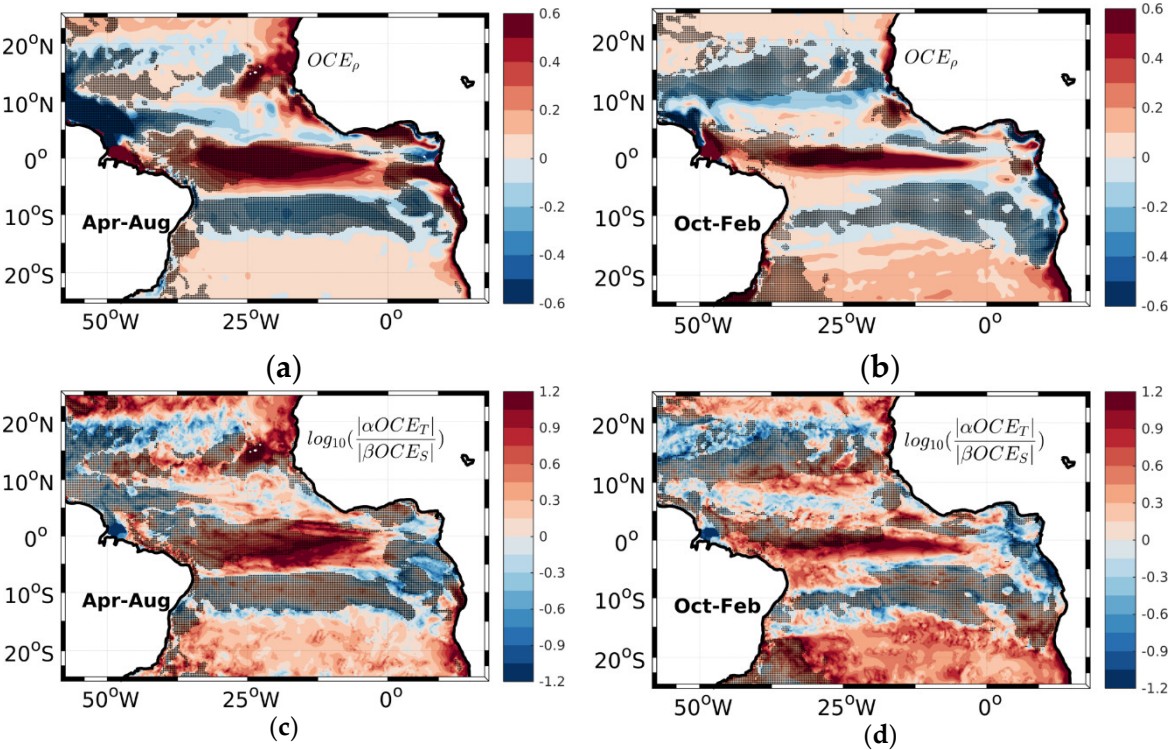

**Figure 4.** Similar to Figure 3 but for the contribution of total oceanic physical processes to mixed layer density variations.

To summarize, the analysis of $OCE_\rho$ first shows a very weak seasonality of this term, and, then, a relatively strong haline contribution around 10° of latitude in both the hemisphere and along the coast. In order to better understand these features, we further decompose the oceanic term into horizontal and vertical contributions.

### 3.3. Horizontal and Vertical Oceanic Physical Processes Contribution

In order to better understand the $OCE_\rho$ effect on the mixed layer density, we split it into oceanic horizontal and vertical processes ($OCEHOR_\rho$ and $OCEVER_\rho$, respectively (Figure 5a,b and Figure 6a,b respectively).

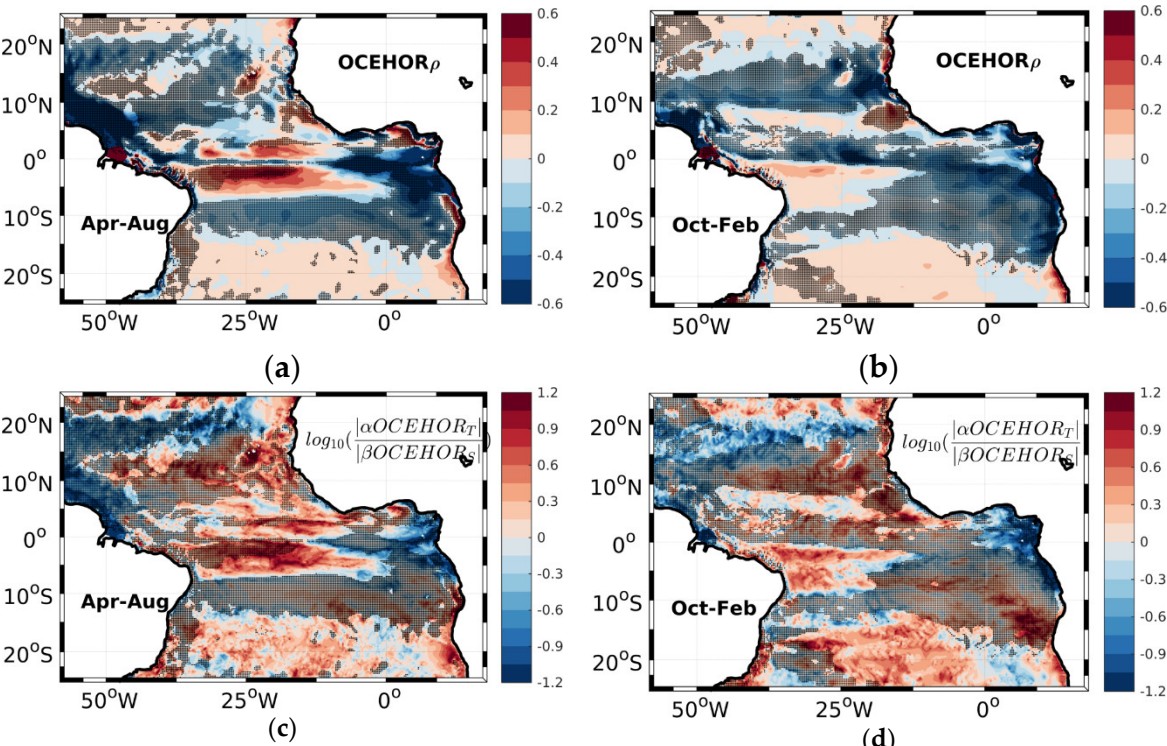

**Figure 5.** April-August (**a**) and October-February (**b**) average of the contribution of total horizontal oceanic physical processes to mixed layer density variations. Units are in kg.m$^{-3}$month$^{-1}$. (**c**,**d**) are the logarithm of the ratio between the absolute value of the total heat and salt horizontal oceanic physical processes contribution. On all panels, regions where heat and salt horizontal physical processes contribution on mixed layer density is the same in terms of signs indicated with black dots.

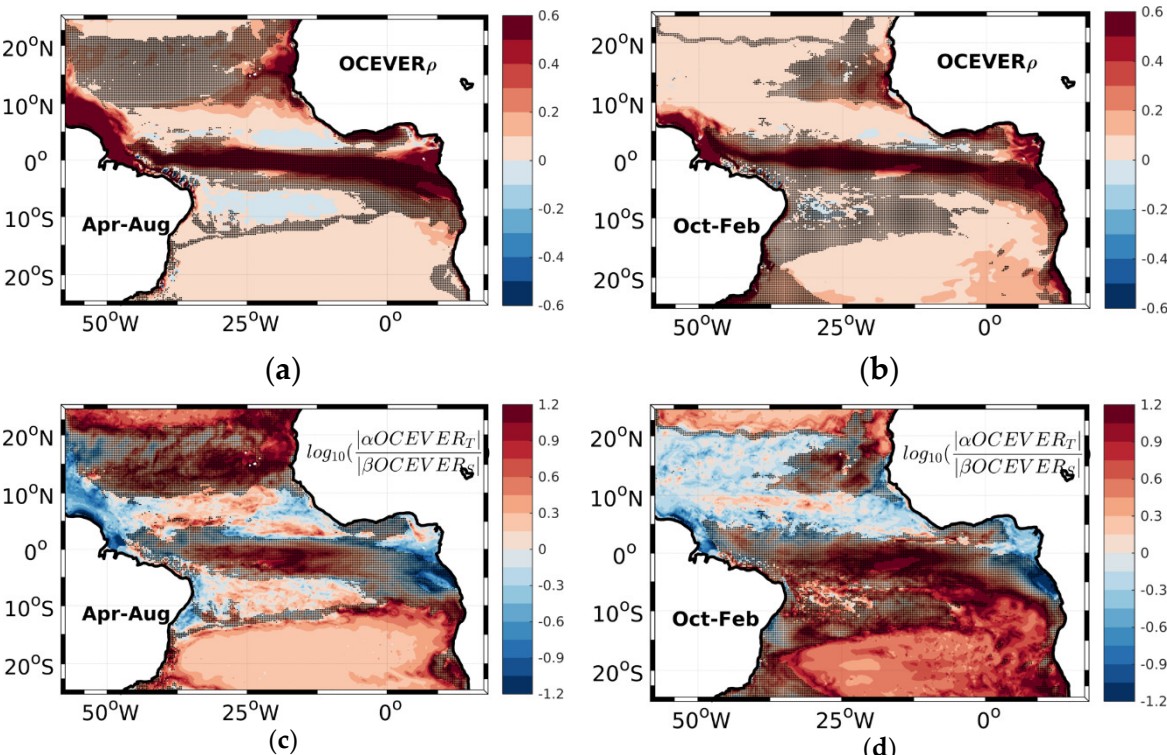

**Figure 6.** Similar to Figure 5, but for the contribution of total vertical oceanic physical processes to mixed layer density variations.

Regarding OCEHOR$_\rho$, one first notices limited seasonality. In general, thermal and haline terms seem to act constructively so as to decrease the density (Figure 5c,d). OCEVER$_\rho$ (Figure 6) has an even more reduced seasonality. It is positive all year long, which means that vertical physical processes increase the mixed layer density throughout the year. In the open ocean, poleward of 15° S and 20° N, the buoyancy decreases by ocean processes (Figure 4) is due to both horizontal and vertical processes. Poleward of 15° S, thermal and haline contributions to both horizontal and vertical processes act destructively with a prevalence of temperature in both cases (Figure 5c,d and Figure 6c,d). Horizontal processes are due to an advection of cold and fresh water mainly from the coast of Angola (not shown). For vertical processes, they increase upper water density by entrainment of cold water while salinity vertical processes decrease the mixed layer density. Poleward of 15° S is an area of maximum salinity so the subsurface waters are fresher than the surface waters due to an excess of evaporation in the subtropical Atlantic. Furthermore, an intrusion flux of salinity will freshen the mixed layer.

Around 10° S and between 10° N–20° N, horizontal processes are the major contributors to the buoyancy increase by ocean processes during both seasons. In addition, temperature and salinity variations act constructively (Figure 5a,b black dots). Thermal and haline horizontal components increase the upper water buoyancy by Ekman advection of warm and freshwater equatorial water, respectively [17]. Between 10° N–20° N, the horizontal haline contribution is at least three times greater than the temperature one (Figure 5c,d). Around 10° S, the thermal component is the main physical process determining the upper density change by horizontal processes. This is mainly due to transport of warm water from the equator by the Ekman transport. At the equator and along Senegal-Mauritanian coast, vertical processes are the major contributor to a buoyancy decrease by ocean processes. This is associated with the intrusion of cold and salty water into the mixed layer by vertical diffusion [17,41,42] with a domination of thermal processes in the equator (Figure 6c,d). This suggests that the vertical gradient of temperature at the mixed layer base is more important for the buoyancy than the salinity along the equator. However, along the Senegal-Mauritanian coast vertical processes, mainly during the boreal autumn-winter, the vertical gradient of salinity at the mixed layer base is more important than the temperature. This could be due to input of freshwater from the Senegal river, which freshen the surface and increases the vertical salinity gradient.

Along the north Brazilian coast and the Gulf of Guinea, horizontal and vertical processes involving salinity are the main contributors to the upper water change by ocean processes with a contribution of at least twice as much as the temperature (Figure 5c,d and Figure 6c,d). In both regions, haline and thermal horizontal processes act together (Figure 5a,b: black dots) due to advection of freshwater from river runoff and warm water from the equator. The fact that haline contribution is largest in the west during the boreal summer could be due to the Amazon discharge peak in May-June [43], which increases the horizontal salinity gradient hence the horizontal advection [17,28]. In these two regions, vertical processes are dominated by vertical diffusion [17,42]. Haline contributions still dominate. Subsurface waters are saltier than in the mixed layer and the vertical diffusion of salinity flux increases its effect on density. On the contrary, subsurface waters are warmer than the mixed layer ones due to temperature inversions in these regions [44] (Figure 7). Consequently, upward temperature flux counteracts the salinity. The prevalence of vertical salinity processes is due to stronger vertical gradient salinity than the temperature at the base of the mixed layer because of freshening of surface water by runoff from Amazon, Niger, and Congo rivers. This is, nevertheless, an important illustration of the impact of a vertical temperature inversion.

Besides the spatial patterns in Section 3.1, Section 3.2, and Section 3.3, we have investigated seasonal evolution of physical processes that drive variations of the mixed layer density (not shown) in the East Atlantic upwelling regions, the equator, and the ITCZ region. Our result illustrates that the density tendency basically keeps the same sign over the two large selected seasons (April–August and October February) primarily by influencing the temperature except in the Angola upwelling and the eastern equator. In the Senegal-Mauritania upwelling (ITCZ region), processes involving salinity are important from September to December (from July to September).

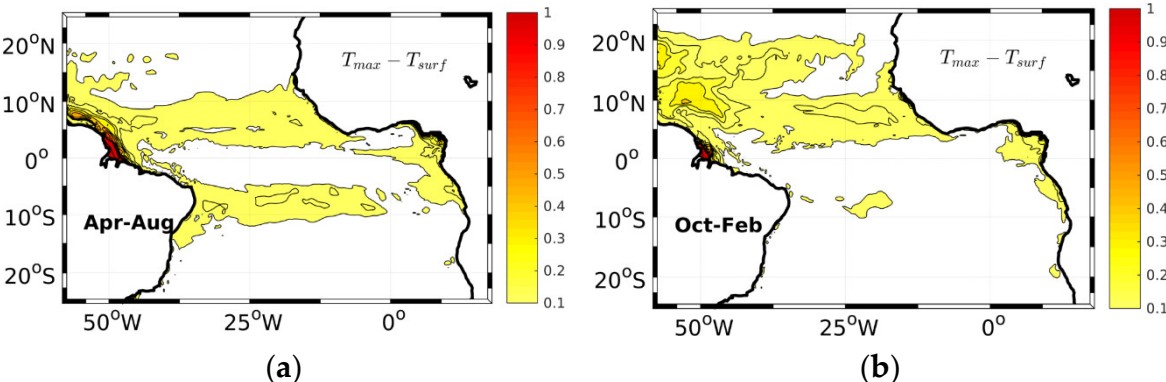

**Figure 7.** April-August (**a**) and October-February (**b**) average of a temperature inversion. Units are in C.

## 4. Conclusions and Discussion

We have investigated, characterized, and quantified the atmospheric and oceanic contributions to mixed layer buoyancy seasonal variation in the tropical Atlantic Ocean. For this, we have used a regional configuration of the NEMO-OPA Ocean General Circulation Model forced with climatological fluxes. This configuration implies a salinity restoring term, which could impact our analysis of the upper ocean processes involving salinity. As discussed in Reference [45], this term is needed in the ocean-only configurations because of the lack of many feedbacks due to the absence of an active atmospheric component and because of the absence of local feedbacks between SSS and freshwater fluxes. These processes can lead to unbounded local salinity trends in response to inaccuracies in precipitation. Several studies have shown that the salinity restoring can strongly affect the oceanic mean state [39], but the impact on the strength of restoring the oceanic mean state is itself not clear and, perhaps, partly model-dependent. For example, while a stronger restoration produces larger transports associated with the Atlantic meridional overturning circulation in Reference [46], the opposite relationship is found in the National Center for Atmospheric Research ( NCAR) model, as discussed in Reference [47]. Our model configuration uses a nominal oceanic resolution of 0.25°. This is not enough to accurately represent the role of meso-scale eddies. These latter have recently been shown to play a fundamental role in transient ocean heat uptake [47] and sea surface spatial distribution [48]. Hence, it plays a role in density. Furthermore, oceanic nonlinear processes such as cabbeling and thermobaricity are found to play an important role in mixed layer density mainly poleward of 30° of latitude [49]. These processes are not taken into account in this simulation. It could be interesting to quantify their contribution in future regional simulations. These processes could also use more accurate forcing data from DFS5.2 or ERA5 to better close the freshwater budget.

Nevertheless, this study has brought insightful information on the upper tropical Atlantic ocean dynamics. Poleward of 10° of latitude, the heat flux mainly controls mixed layer buoyancy, which is at least three times greater than the freshwater flux whatever the season. During boreal spring-summer of each hemisphere, the freshwater flux partly compensates the dominating heat flux in terms of buoyancy loss while, during the fall-winter, they act together to increase the mixed layer density. The seasonal compensation effect is due to the dominance of shortwave radiation in the spring-summer while freshwater flux is dominated by evaporation throughout the year. In major open ocean regions, the atmospheric buoyancy fluxes dominate over the oceanic contribution to set up the mixed layer density year-round. The seasonality of the oceanic processes is much less marked than the atmospheric ones. All year long, they contribute to increase density in the equatorial region and poleward of ~20° of latitude, and they reduce density around 15° N and S. This is an important outcome of our study, which is important for a good understanding of the interplay between the upper ocean layer and subsurface ocean layer.

The weak seasonality of oceanic processes is due to both horizontal and vertical oceanic processes. The horizontal processes generally decrease the mixed layer density with strong contributions in the vicinity of rivers' mouths (Amazon and Congo rivers). Horizontal processes involving salinity are more important around 15°N of latitude and around the rivers' mouth. In the latter area, the salinity processes are at least twice more than the first temperature mainly due to input of freshwater from the rivers. In these regions dominated by salinity, temperature and salinity processes involved in horizontal advection act generally together due to mainly a transport of warm and fresh water. Concerning vertical processes, they generally increase the mixed layer density with a strong contribution along the equator. This is mainly due to vertical diffusion, which includes the major vertical oceanic processes. Generally, in temperature inversion regions, salinity and temperature processes involving vertical processes compensate each other while, in other regions such as equatorial and coastal upswelling, they act together. Temperature processes are generally the major contributor in vertical processes mainly during the Spring-Summer except in the Gulf of Guinea, the Senegal-Mauritania upswelling, and the north Brasilian coast where the salinity processes are the main contributors. The dominance of temperature processes is mainly due to a stronger vertical temperature gradient at the base of the mixed layer when compared to the salinity vertical gradient. In the regions cited above, however, the vertical salinity gradient takes over because of strong freshwater input.

**Author Contributions:** I.C., J.M., N.K., and A.L. conceived the experiments and the methodology. Software, I.C. Validation, I.C., J.M., and N.K. Formal analysis, I.C., J.M., N.K., A.L., and T.L. Writing-original draft preparation, I.C., J.M., N.K., A.L., and T.L. Writing—review and editing, J.M., N.K., A.L., and T.L. Funding acquisition, J.M. and A.L. All authors have read and agreed to the published version of the manuscript.

**Funding:** IRD, grant number 182SNLMIE2, funded this research. This study was also supported by the CNES TOSCA SMOS OCEAN project.

**Acknowledgments:** The authors thank the Laboratoire de Physique de l'Atmosphère et de l'Océan Siméon Fongang (LPAO-SF) of the University Cheikh Anta Diop de Dakar where the work was mainly done. The editor and the anonymous reviewers are also sincerely thanked for their valuable comments and suggestions to improve the quality of the paper.

**Conflicts of Interest:** The authors declare no conflict of interest.

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
