# Peer review of "Tropical Atlantic Mixed Layer Buoyancy Seasonality: Atmospheric and Oceanic Physical Processes Contributions"

_atmosphere, doi:10.3390/atmos11060649_

Round 1

Reviewer 1 Report

Major issues

The incompleteness of the model equations description, in addition to the graphical and notation problems, affecting Section 2.2 make the manuscript difficult to read. Section 2.2 must be expanded and improved to allow the reader to fully understand the model equations and then the following sections of the paper.

In Section 2.3 is not presented an evaluation or validation exercise, but rather a graphical comparison. It is necessary to clarify this issue in the text and to appropriately modify the terminology.

Summarizing, Section 2 is the critical one, while the rest of the paper is well written.

Minor issues

Line 23: “Inter-tropical Convergence Zone” instead of “ITCZ”.

Line 30: keywords should be more concise.

Lines 35-37: to check the contents.

Line 77: to describe the OPA meaning.

Lines 77-78, 87: the nomenclature should be harmonised.

Section 2.2: there are graphical problems about the numbering of equations.

Eq. 1 and line 110: to check the symbols, to describe the mld meaning.

Line 116: to describe the ml meaning (see line 119).

Line 119-127: all the symbols in the equations have to be defined (E? P? R? and so on).

Line 128: to check the equations numbers.

Lines 131-133: the sentence is not clear.

Line 134: the parenthesis notation has to be harmonised.

Lines 136-142: it is very difficult to follow the drift, due to a number of equations typos, I guess.

Section 2.3: the graphical and notation problems affecting equations of Section 2.2 inevitably make unclear this section. In any case I suggest to write the equations in the figures using larger characters, mainly in Figure 2.

Line 167: the restoring term concept should be introduced in Section 2.2.

Line 197: I can’t see the dots in the figures, do you mean shaded may be? See also line 191 e line 229. To harmonize nomenclature.

Line 208: 3d instead of d.

Line 321: to check the figure caption.

Lines 323-325: these summary lines should be more informative: where? which model?

Ref. 28: to check the title.

Reviewer 2 Report

The paper investigates the physical processes controlling the mixed layer buoyancy by analyzing the mixed-layer heat and salt budget using a linealyzed equation of state. The methodology is interesting and new results are presented. However, I think that the paper needs some extensive rewriting and recommend to send back to the authors for a major revision

Major comments

  1. Could you, please, justify the use of surface values for validation? Why do not use a reanalysis
  2. Besides the spatial patterns, would be interesting to see the seasonal evolution of the terms in selected locatios, as the bands around 10o, the equator, the East Atlantic upwelling region.
  3. How will the results influenced by the atmospheric forcing? How about using DRAKKAR DFS5 or ERA5?
  4. I wonder if ocanic nonlinear processes as cabbeling should also taken in account
  5. Please comment on the impact of SST restoring, if any on the results and what do you expect in coupled run. Again, using a n oceanic reanalysis should not be more useful?

Minor comments:

  1. Line 36. The phrase “introduction should briefly place the study in a broad context and highlight why it is important” looks strange   
  2. Line 41-43: what about advection of subsurface salty water?
  3. Line 76: “gaining understanding of the physical mechanisms driving”
  4. Line 87: ” For this study we use the OGCM-OPA oceanic model “
  5. Line 96: probably you wanted to say “global 1/4o
  6. “depends non-linearly”
  7. Line 117; replace “Both approximation made above will be verified (but not show) in the model and in validation data.” by “Both approximations will be verified (but not shown) for model and validation data”
  8. Would be nice to have an estimate of the error of the linearization for the barrier layers
  9. I do not understand why the diffusion is considered as oceanic (OCE). It seems to be considered more important than vertical (OCEVER) and horizontal (OCEHOR) effects. 
  10. Lines 191-193 could be rewriting to make them more clear
  11. Line 310: which role plays the representation of the river discharge in the model for these results?
  12. Line 220: precipitation and runoff

Reviewer 3 Report

..

Round 2

Reviewer 2 Report

The authors have responded satisfactorily to many of my questions, but some responses are not completly atisfactory for me.

Question 1. I was asking why they use surface values for evaluation of mixed layer quantities. Now they make a valid point, but I would like to see if the results hold with another, "coupled" reanalys (e.g. CFSR)

question 5. I was truly surprised by the answer to my 5th major comment. The authors say in their answer "In this study, we use neither SST
restoring nor a coupled run" However, in the manuscript in lines 105 and 106 of the revised manuscript they rfer to a "newtonian restoring term". Besides, as it is well known, the coupling introducs new, non-trivial biases in the model, and conseqwuently, should influence their results 

Author Response

Revisions round2 of paper #atmosphere-786707 by CAMARA et al. Dear editor. Please find in attached file our detailed response to the reviewer 2 comments. All points were  addresses, we are grateful the reviewer for their detailed comments, which helped improve the   manuscript.

Reviewer 3 Report

It is recommended to publish this work as is.

Author Response

Dear Reviewer,

Thank you very much.

Round 3

Reviewer 2 Report

The authors again gave a superficial answer to my questions.

It was never

"a confusion for the reviewer" 

however, comparing a newton restored model with an assimilated reanalysis raises question about how much the related non-physical forcings influence the results. With EN4 the results seem to be robust enough. 

As the results could be relevant for coupled models, I asked about them. I would like to hear what thoughs the authors on this regard.

As the paper now looks good enough, I recommend it for publication after they answer to my question 

Author Response

Revisions round3 of paper #atmosphere-786707 by CAMARA et al. Dear editor. Please find below our detailed response to the reviewer 2 comments. All points were addresses, we are grateful the reviewer for their detailed comments, which helped improve the manuscript.

Comments and Suggestions for Authors

The authors again gave a superficial answer to my questions.

It was never

"a confusion for the reviewer"

however, comparing a newton restored model with an assimilated reanalysis raises question about how much the related non-physical forcings influence the results. With EN4 the results seem to be robust enough. As the results could be relevant for coupled models, I asked about them. I would like to hear what thoughs the authors on this regard. As the paper now looks good enough, I recommend it for publication after they answer to my question

We apologize since we did not understand correctly the reviewer’s remark. As we understand it now, the reviewer asks how our results would be transposed to climate models. Indeed the latter are biased (GRODSKY et al., 2011; Doi et al., 2012...), with typically too warm equatorial waters (underestimated equatorial upwelling) and underestimated salinity minima in the vicinity of major rivers (Griffies et al. 2005) due to partially unphysical restoring term. Both these biases, can affecte the ocean density, and the ocean stability and vertical mixing are altered accordingly, thus leading to the vertical adjustments of the salinity and temperature. Such local modifications can spread from the original forcing region to other basins through dynamic ocean processes, thereby leading to the perturbation of salinity and temperature in areas far from the forcing region.

As a consequence of salinity minima underestimation in the vicinity of river, it could decrease the vertical stratification and such lead to depen the halocline hence the mixed layer. It could also, decrease the barrier layer thickness then increase subsurface warm water intrusion in western boundary mixed layer thus influencing air-sea interaction. According to Wu and Sun, [2010], a salinification (due to large difference between the model and reference salinity) in the western tropical Pacific reduces the sea surface height locally and sets up a positive zonal pressure gradient anomaly, inducing anomalous poleward surface flow and in turn subsurface equatorward-compensating flow.

Despide the biases found in coupled and no coupled model, it is important to point out that these models allow us to better understand the climate. However, there is a need to improve their accuracy by using new forcing data (e.g. DFS5.2), improve spatial resolution, use alternative approaches to close freshwater and heat budgets.

Round 4

Reviewer 2 Report

The paper can be published in its present form